# Bots, Elections, and Controversies:
# Twitter Insights from Brazil's Polarised Elections

## ABSTRACT

From 2018 to 2023, Brazil experienced its most fiercely contested elections in history, resulting in the election of far-right candidate Jair Bolsonaro followed by the left-wing, Lula da Silva. This period was marked by a murder attempt, a coup attempt, the pandemic, and a plethora of conspiracy theories and controversies. This paper analyses 437 million tweets originating from 13 million accounts associated with Brazilian politics during these two presidential election cycles. We focus on accounts' behavioural patterns. We noted a quasi-monotonic escalation in bot engagement, marked by notable surges both during COVID-19 and in the aftermath of the 2022 election. The data revealed a strong correlation between bot engagement and the number of replies during a single day ($r = 0.66$, $p < 0.01$). Furthermore, we identified a range of suspicious activities, including an unusually high number of accounts being created on the same day, with some days witnessing over 20,000 new accounts and super-prolific accounts generating close to 100,000 tweets. Lastly, we uncovered a sprawling network of accounts sharing Twitter handles, with a select few managing to utilise more than 100 distinct handles. This work can be instrumental in dismantling coordinated campaigns and offer valuable insights for the enhancement of bot detection algorithms.

## CCS CONCEPTS

• **Do Not Use This Code → Generate the Correct Terms for Your Paper**; *Generate the Correct Terms for Your Paper*; Generate the Correct Terms for Your Paper; Generate the Correct Terms for Your Paper.

## KEYWORDS

Brazilian Elections, Bots, Twitter, Political Networks, Polarisation

**ACM Reference Format:**
Anonymous Author(s). 2024. Bots, Elections, and Controversies: Twitter Insights from Brazil's Polarised Elections. In *Proceedings of Companion Proceedings of the Web Conference 2024 (WWW '24 Companion)*. ACM, New York, NY, USA, 8 pages. https://doi.org/XXXXXXX.XXXXXXX

## 1 INTRODUCTION

Brexit in Europe, Trump in the U.S., and Bolsonaro in Brazil exemplify the escalating polarisation characterising political discourse worldwide [13]. Simultaneously, the pivotal role of online social

platforms as primary mediums for campaigns, debates, and recruitment has come to the forefront [4, 46, 47].

The presence of bots in electoral campaigns has seen a year-on-year increase, coinciding with a growing academic focus [6]. This expanding body of literature delves into elections worldwide, encompassing the 2016 U.S. elections [5], the 2017 electoral contests in Germany [20] and in France [15], Italy in 2018 [35], Spain in 2019 [32], elections across numerous African countries during 2017-2018 [29], the Asia-Pacific region in 2019-2020 [44], and the 2019 European Parliament elections [33], to name a few. However, a more pressing concern emerges as the diffusion of misinformation disproportionately affects accounts depending on their political affiliations [5, 8, 22].

In 2016, Brazil experienced political turmoil with the impeachment of Dilma Rousseff. Subsequently, the years 2018 and 2022 witnessed Brazil's most hotly contested elections in its history, culminating in the elections of far-right candidate Jair Bolsonaro, followed by the left-wing figure Lula da Silva. This era was overshadowed by a murder attempt, a coup attempt, the pandemic, and a profusion of conspiracy theories and controversies, creating fertile ground for misinformation. Notably, Brazilian datasets have been at the forefront of developing computational methods for detecting propaganda [2], countering misinformation in advertisements [40], identifying low-credibility Brazilian websites [11], and fact-checking images [36]. WhatsApp groups, immensely popular in Brazil, played a pivotal role in monitoring misinformation spread during the 2018 elections [24]. Furthermore, substantial criticism has emerged regarding the use of misinformation as a political weapon during the COVID-19 pandemic [37] and culminating in Bolsonaro's ineligibility until 2030 [39].

In this study, we harness social media data and network analysis to discern and illuminate population-level political behaviour in Brazil. Our analysis tracks the evolution of political groups, from contentious competitors during campaigns to government and opposition blocks after elections. Our findings illuminate a transition from a pre-election phase marked by numerous polarised groups to a post-election phase in which these factions coalesce into government and opposition clusters. Our investigation uncovers a sprawling network of coordinated accounts that share Twitter handles. We also observe a pronounced surge in bot engagement, with noteworthy peaks during the pandemic and in the aftermath of the 2022 election. Furthermore, our data underscores a strong correlation between bot engagement and the number of replies. Finally, we identify anomalous days characterised by an unexpectedly high number of account creations.

We employed the Twitter streaming API to monitor fourteen Brazilian presidential candidates during the 2018 elections, and thirteen candidates and twenty-seven political parties during the 2022 cycle. The data collection spanned from August 30, 2018, to March 14, 2023. The period encompasses 1,657 days, and the collection

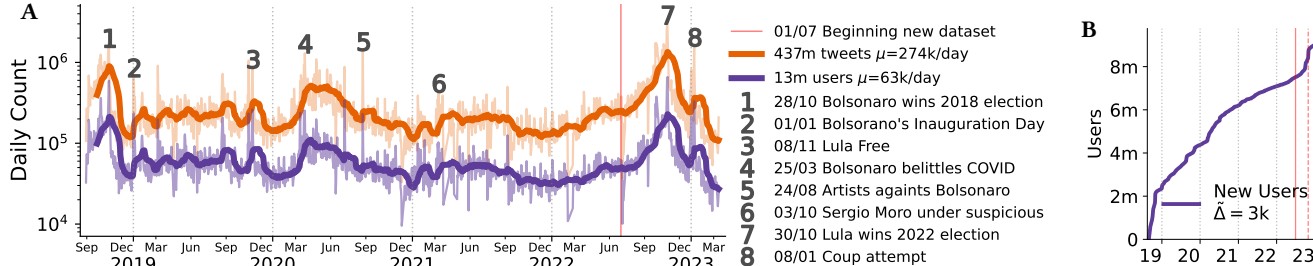

Figure 1: A Political Tale from Tweets — (A) Timelines spanning two election cycles (2018/22), covering 1657 days and involving 437 million tweets (in orange) from 13 million accounts (in purple). Daily tweet counts are represented by the lighter lines, while the 30-day moving average is depicted in bold. Key events, such as the 2018 (#1) and 2022 (#7) election days, are highlighted for reference. (B) The cumulative plot illustrates a continuous increase in the number of distinct accounts joining the political conversation for the first time. The red vertical lines (1/July/22) indicate the beginning of the 2022 election cycle, while the red dashed line marks the election day.

process remained active for 94% of this time. This comprehensive effort resulted in the acquisition of a vast dataset comprising 437 million tweets originating from 13 million distinct accounts.

## 2 DISSECTING TWITTER ACCOUNTS

### 2.1 Dynamics of political engagement

The Twitter timeline depicted in Figure 1A reveals discernible shifts in political engagement. In 2018, there is a notable surge in activity leading up to the election day, followed by a decline in the period between the release of election results and the inauguration day (January 1, 2019). Subsequently, the volume of tweets and active users stabilises, punctuated by occasional peaks corresponding to significant events. The volume of tweets remained relatively low throughout 2020 until the onset of COVID. Subsequently, a series of peaks emerged, driven by discussions surrounding both the pandemic and political developments. The most significant surge occurred at the beginning of 2022, building steadily until the election day. A pattern akin to 2018 repeats as there is a decline between the election and the inauguration. Notably, 2022 also witnessed an abrupt surge coinciding with the coup attempt on January 8, 2023.

Despite the somewhat consistent daily number of accounts engaging in the conversation, Figure 1B reveals an intriguing trend wherein more than three thousand new accounts join the Brazilian political discourse each day. This observation hints at an account churn rate of approximately 5%. Importantly, the introduction of new terms into the data collection on July 1, 2022, does not appear to have significantly influenced the influx of new accounts. In forthcoming research endeavours, we intend to delve deeper into the dynamics of accounts exiting the conversation. One plausible interpretation for this is that it may be driven by a substantial presence of bots within the Twitter ecosystem. These bots could potentially be replaced by new ones as they are suspended by the platform for policy violations. However, it is essential to note that in our current analysis, we did not assess bot activity among the incoming accounts.

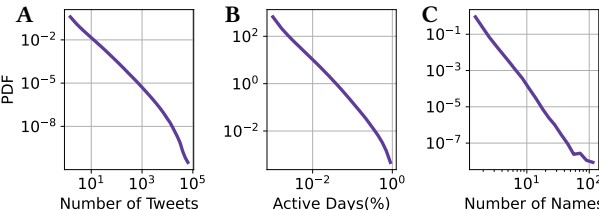

Figure 2: Heterogeneous Behaviour of Accounts — (A) Distribution of the number of tweets per account, showcasing a long-tail pattern where a small number of accounts post close to 100k tweets. (B) Distribution of the number of active days, revealing exceptional accounts that tweet every day. (C) Distribution of the number of distinct screen names used by accounts, with some accounts utilizing more than 100 different names.

### 2.2 Accounts' heterogeneous characteristics

Consistent with observations in various social systems, our dataset underscores the presence of accounts exhibiting heavy-tailed properties. Figure 2 illustrates this phenomenon, wherein the majority of accounts contribute relatively few tweets, while a select few manage to produce an exceptionally high volume, nearing 100,000 tweets within the specified timeframe. It is worth noting that, despite Twitter's imposed limit of 2,400 tweets per day, some accounts employ strategies to circumvent this restriction, often through the adoption of abusive deletion behaviours [43].

The skewness observed in the distribution of tweet volume is mirrored in the distribution of active days. While the majority of accounts engage for just a few days, there exists a subset of accounts that remain active on a daily basis. However, it is imperative to acknowledge that the figures presented herein may be underrepresented, as they pertain exclusively to the tweets captured by our data collection. These extremes in user behaviour raise suspicions of automation.

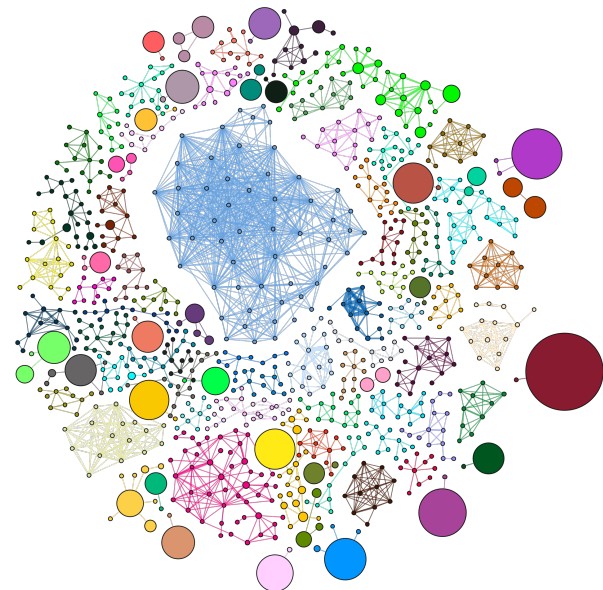

Figure 3: Screen Name Sharing Network — Each node in the network represents a Twitter account, and they are connected if they share a common handle (*screen_name*). The size of each node is proportional to the number of tweets posted, and different colours represent various suspicious coordinated groups (connected components). For clarity, we display only groups consisting of at least 10 accounts or those responsible for producing more than 10,000 tweets.

Prior research has highlighted the association of multiple handles (i.e., screen names) used by a single account or shared among multiple accounts with potentially malicious activities [25] and coordinated campaigns [31]. Figure 2 further elucidates this trend by illustrating the distribution of the number of distinct names employed by the accounts within our dataset. It is noteworthy that some accounts exhibit the use of more than a hundred distinct handles, amplifying concerns of potentially deceptive practices.

## 2.3 Coordinated accounts

Pacheco et al. [31] introduced a framework for identifying coordinated campaigns on Twitter, focusing on the presence of shared handles among multiple accounts . This entails different accounts, signified by distinct user_id, adopting the same perceived identity, denoted by identical screen_name. Importantly, this methodology enables the detection of coordinated groups of accounts, regardless of their automation level, extending the scope beyond bots.

Figure 3 displays the outcomes of the coordination detection [31] within our dataset. The figure unveils numerous connected components, representing the coordinated groups, which vary in size. Notably, the figure emphasises the most suspicious groups, filtered either due to their size, with more than ten accounts involved, or because of their prolific engagement in the conversation, generating over ten thousand tweets.

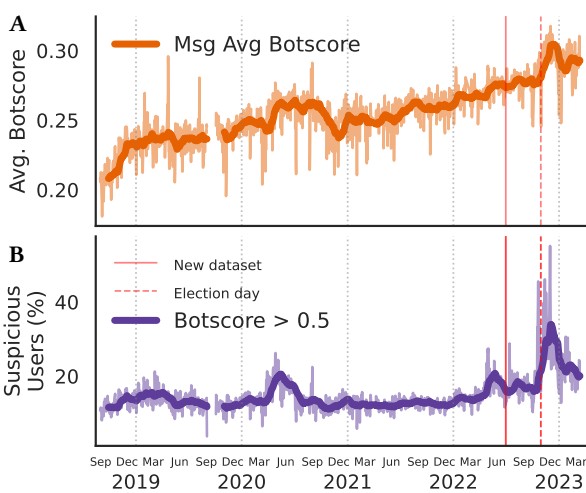

Figure 4: Increasing Bot Activity — (A) The daily average botscore derived from tweets exhibits a continuous upward trend, reaching peaks in the days following the 2022 election. (B) The percentage of accounts exhibiting bot-like behaviour remains relatively stable in the dataset, with notable increases observed after the initial wave of the pandemic and following the 2022 election.

It is important to highlight that, in this study, we did not uncover groups involved in name squatting or hijacking, as previously reported in the literature [25, 31]. This discrepancy can be attributed to the distinctions between our dataset, which is domain-specific, and the datasets employed in prior research, which were domain-agnostic. Furthermore, our analysis refrains from delving into the specifics of the campaigns undertaken by these groups or their overall impact on the broader discourse. These facets will be addressed in future research.

## 2.4 Bots engagement

In this section, we explore the temporal evolution of bot engagement by utilising BotometerLite [49], a tool designed for the assessment of bot-like activities within social media data. It is essential to acknowledge that while bot detection algorithms are valuable, they are far from infallible [12]. These algorithms have faced criticism on various fronts, including concerns about their lack of transparency [16], the presence of elevated numbers of false negatives and false positives [18, 28], and issues of historical data [9]. To mitigate some of these criticisms, our analysis focuses on bot activity as a broad trend, avoiding specific account-level scrutiny or rigid threshold definitions.

BotometerLite [49] operates by assessing a single tweet, specifically the *user profile object* within a tweet, to assign a *botscore* to the account responsible for that tweet. The botscore, which can range from zero to one, serves as an indicator of the extent to which an account's features resemble those of a human versus automated account (bot) activities. It is important to note that the *user profile features* used for this assessment are subject to change over time,

meaning that even two consecutive tweets from the same account may yield different botscores.

For our analysis, we define the *daily botscore* of an account as the average botscore derived from all of its tweets within a given day. Additionally, we establish the concept of *bot engagement* or *content botscore*, denoting the average botscore calculated from all tweets collectively.

As illustrated in Figure 4A, we present the evolving landscape of bot engagement within the discourse surrounding Brazilian politics. While the daily engagement displays noticeable fluctuations, the moving average reveals a pronounced upward trajectory that has persisted since the commencement of our data collection in 2018. Notably, we observe a significant surge in bot engagement commencing in March 2020 during the pandemic, and this trend further intensifies in the aftermath of the 2022 elections. This observed trend aligns with findings reported by academics and media outlets, which have highlighted the escalating dissemination of disinformation in Brazil. Of particular concern are unsubstantiated claims, often attributed to Bolsonaro, regarding the e-voting system [7, 17, 38].

The recent acquisition of Twitter has sparked considerable controversy regarding the prevalence of bots on the platform [21]. In 2017, Varol et al. [45] proposed a method for conducting a census of Twitter accounts and found that approximately 9% to 15% of accounts were likely to be social bots. In this study, we refrain from providing a specific numerical estimate and instead examine trends surrounding the presence of bots within our dataset.

Figure 4B portrays the temporal evolution of the daily percentage of what we term *suspicious users*. In our analysis, we categorise *suspicious* accounts as those with a botscore exceeding 0.5. The percentage of bots within this category remains relatively stable, fluctuating between 15% and 20%. It is worth noting that varying thresholds for suspicious accounts would result in different quantities of bots, but the overall stable trend persists. Noteworthy spikes in bot activity occur during COVID and in the aftermath of the 2022 elections, with specific days registering a particularly high proportion, exceeding 50% of the total accounts.

The convergence of two key results, namely the escalating content botscore and the sustained proportion of bots, offers compelling evidence that bots are progressively intensifying their involvement in the ongoing conversations. This observation raises important questions about the effectiveness of existing measures aimed at countering and mitigating bot activities. It hints at the possibility that current initiatives designed to combat and block bots may not be sufficient to curtail their presence and influence.

## 2.5 Replying bots

Mbona and Eloff [27] employed a combination of Benford's Law, Principal Component Analysis (PCA), and random forest techniques to identify discriminative features for bot detection. Notably, their findings underscored that the number of retweets serves as an effective discriminator, whereas the number of replies did not exhibit the same discriminatory power. In contrast, Pozzana and Ferrara [34] demonstrated that both the fraction of retweets and replies tend to be more prevalent in human interactions compared to bot-driven activities. Finally, Mazza et al. [26] distinguished between trolls and

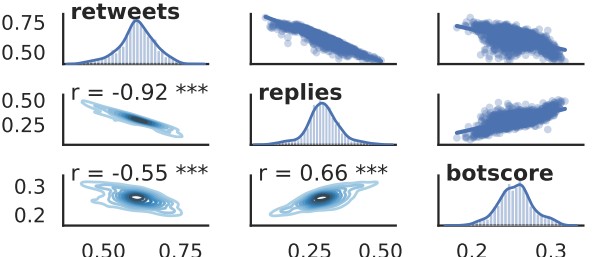

**Figure 5: Relationship Between Retweet and Reply, and Bot Engagement** — The percentage of *replies* in a day exhibits a significant positive correlation ($r = 0.66$) with bot engagement, while the percentage of *retweets* demonstrates a negative correlation ($r = -0.55$) with bot activity.

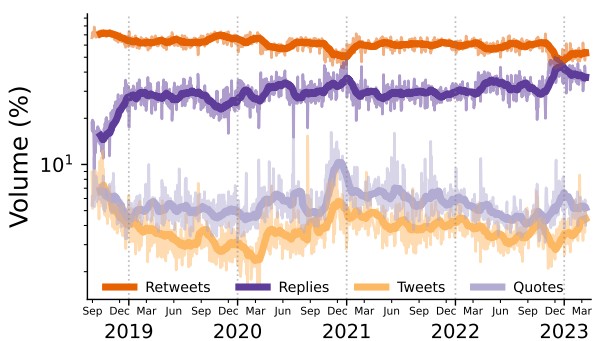

**Figure 6: Evolution of Tweet Types Over Time** — While *retweets* remain the dominant type of tweets, there is a noticeable increase in the popularity of *replies* over time.

social bots, revealing that the latter tend to employ a higher volume of replies than human users.

In this section, we delve into the intricate relationships among these engagement metrics and the overarching *content botscore*, as defined in Section 2.4. A lower content botscore signifies a scenario in which the majority of tweets originate from accounts that exhibit human-like characteristics, while a higher content botscore indicates a greater involvement of automated accounts in the conversation. Figure 5 showcases the distributions and correlation patterns among the percentage of retweets, percentage of replies, and the content botscore. Our results shed light on the distinct tendencies of bots, particularly their propensity for engaging through replies, offering valuable insights into the interplay of these engagement metrics.

Figure 6 offers a chronological perspective on the proportions of different tweet types. Throughout this timeline, retweets consistently dominate the landscape, maintaining a prominent presence. However, notable shifts in tweet composition are observed. For instance, there is a discernible 10% decline in the retweet rate, plummeting from 71% prior to the second round of the 2018 election to 61% following the inauguration day. In stark contrast, the number of replies more than doubled during the same period, surging from

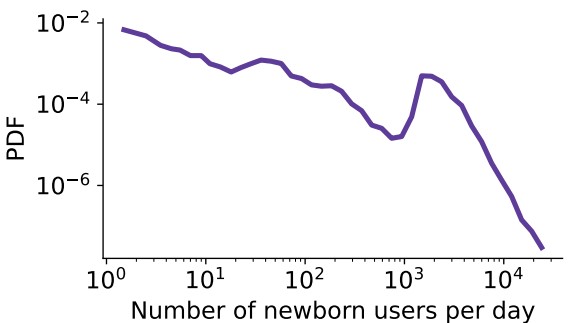

**Figure 7: The distribution of the number of accounts created per day.**

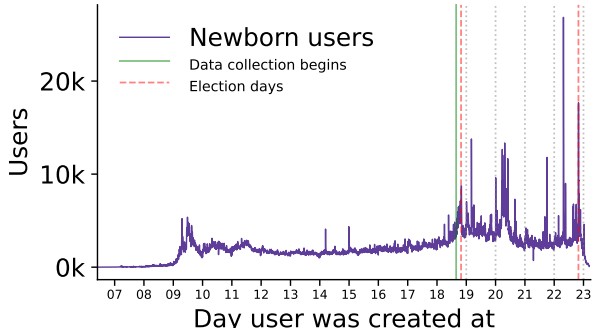

**Figure 8: Accounts Created on the Same Date ("Birthday Twins") — This figure displays the count of accounts created per day, highlighting the age diversity of Twitter accounts in the dataset. Unusual peaks are also observed, including accounts created as far back as the early days of Twitter.**

14% to 30%. These changes may reflect two distinct behavioural patterns: the prevalence of propaganda-oriented content during election campaigns, juxtaposed with an emphasis on discourse and debate in the post-election mandate period. This phenomenon, characterised by shifts in the composition of tweet types, was not unique to the 2018 election cycle but recurred during the 2022 cycle as well at a lower scale.

## 2.6 Uncovering accounts "birthdays"

Contrary to the zodiac, online "dates of birth" can reveal a wealth of information beyond just an account's age. Prior research, such as Tardelli et al. [42], has demonstrated that financial social bots often share similar creation dates. Jones [19] successfully detected bots exploiting the Gulf crisis primarily by analysing account creation dates. Similarly, Takacs and McCulloh [41] utilised creation dates to identify dormant bots during the 2018 US Senate election. These bots [41] were not particularly active, yet they attempted to exert influence based on their substantial number of followers.

In our investigation, we embark on a quest for days marked by a substantial influx of "newborn" accounts. We scrutinised the creation dates of each of the 13 million accounts actively participating in the discourse surrounding Brazilian politics and categorised them accordingly. Figure 7 visually depicts the distribution of the number of "newborns" per day. This distribution exhibits a fat-tailed pattern, characterised by a decline in the probability of multiple accounts being created on the same day up to around 1,000 creations daily. Subsequently, the probability rises, peaking at approximately 3,000 accounts per day, before sharply declining once more.

Figure 8 provides a timeline of account creation counts for those accounts born on the same date. Notably, the plot expectedly reveals that numerous accounts were established long before our data collection began, with some dating back to the inception of Twitter itself. However, the plot also unveils peculiar and anomalous peaks, primarily concentrated in 2020, with a maximum day in 2022 registering the creation of over 20,000 accounts.

Of particular note is an outlier peak on 1st January 1970, which we chose to omit from the plot. This anomaly, marked by the creation of 41 accounts on a date preceding the existence of the Twitter platform itself, is unequivocally suspicious. The existence of accounts with creation dates preceding the platform's inception, as well as those exhibiting multiple creation dates, presents a puzzling phenomenon. We remain uncertain whether this issue is an innocuous glitch or a deliberately orchestrated malicious activity. Although we have not encountered official reports on this matter, it has garnered attention on social media [30].

In future research, we plan to delve deeper into the analysis and characterisation of these enigmatic accounts, shedding light on their origins and potential significance.

## 3 DISCUSSIONS

*Bot Detection Challenges.* The significant increase in bot engagement, as evidenced by our findings, underscores the escalating concerns about our capacity to effectively combat fringe actors. Our use of BotometerLite, reliant on historical data and a model trained in 2020, might not fully encapsulate the evolving nature of bot behaviour, especially during critical events like elections. Research has illuminated the adaptive tactics of bots during elections and the formidable challenges in detecting automated accounts [23]. Detecting bots has become a more intricate task, and the proliferation of misinformation may be greatly exacerbated by innovations like GPT [14, 48].

*Dataset Bias.* While our analysis is grounded in a substantial sample of online users, it remains uncertain how representative Twitter data is of the broader Brazilian political spectrum. It is crucial to acknowledge that no dataset is entirely free from bias. Many research efforts rely on datasets constructed using dynamic keyword-based approaches, which involve continually updating tracking terms to adapt to the evolving online environment. For example, some researchers employ snowball techniques to harvest new hashtags, resulting in datasets that are tailored to current trends. In contrast, our approach was distinct. We aimed to minimise changes to tracking terms. For instance, we retained hashtags primarily associated with campaign periods throughout our study duration (see Tables 1 and 2). Similarly, we continued tracking all presidential candidates

even after the elections. Notably, a substantial proportion of these candidates remained actively engaged within the Brazilian political landscape. A striking 46% of the 2022 candidates were participants in the 2018 cycle. Some of these candidates joined coalitions to support new contenders, while others assumed leadership roles in political parties or government. Maintaining a stable list of tracked individuals allowed us to consistently monitor Brazilian politics without introducing additional bias stemming from trending topics.

*Twitter's Evolution.* The transformation of Twitter has gone beyond a mere re-branding to $X$. The new API introduces significant limitations on data collection, which have the potential to hamper the monitoring capacity of academics and open new avenues for exploitation by malicious actors. However, it is not yet clear whether certain behaviours observed on "old" Twitter have ceased to exist on $X$. Consequently, it is imperative to continue exploring datasets from the older version of Twitter. Furthermore, it's unlikely that researchers can reconstruct such a comprehensive dataset as the one presented here. Although we are unable to directly share our data, we are actively seeking collaborations to expand and extend this research.

*Future Research Directions.* Future work should delve into the dynamics of accounts joining the political discourse. Who constitutes the persistent core of participants? Does the churn rate only capture isolated instances of engagement? Do accounts engage periodically or based on specific topics? The coordinated suspicious groups and accounts created on the same day warrant further investigation, including characterisation efforts to identify who these actors are and the subjects they discuss. Additionally, it is imperative to measure the impact of their actions on the overall conversation and trace back groups involved in the coup attempt. Despite the lingering questions, we anticipate that this work will play a pivotal role in dismantling coordinated campaigns and offer valuable insights to enhance bot detection algorithms.

In summary, our research has revealed the alarming growth in bot engagement, raising concerns about our ability to combat fringe actors effectively. While our study is not without limitations, such as the evolving nature of bot behaviour and dataset biases, it has provided valuable insights into the landscape of Brazilian politics. As we confront the challenges of evolving social media platforms and advancing technologies, it is imperative to continue probing these issues and collaborating to develop effective solutions.

## 4 DATA COLLECTION AND CONTEXT

This paper examines a dataset consisting of 437 million tweets generated by 13 million accounts associated with Brazilian politics between 2018 and 2023. Before delving into the specifics of data collection, it is essential to provide a contextual overview of the Brazilian electoral process.

Brazil operates as a federal presidential representative democratic republic with a multi-party system, comprising 27 federal units (states and a federal district). Voting in Brazil is mandatory for individuals aged between 18 and 70 years, while it is optional for those under 18, over 16, or over 70. Elected Brazilian politicians generally serve four-year terms, and the population is required to select their representatives in general elections every two years,

alternating between federal and local elections. For instance, the years 2018 and 2022 constituted federal elections for the positions of president, governor, and federal congressmen, while 2016 and 2020 featured elections for mayor, state deputies, and city councillors. Elections in Brazil are conducted on a single day, during which all votes must be cast in person, typically on a Sunday in October, between 8 AM and 5 PM. In cases where no candidate secures an absolute majority of the valid votes (more than 50%), a second round of voting is held, featuring the two leading candidates.

Since 1996, Brazil has employed electronic voting machines, which have eliminated paper-based fraud and enabled rapid result tabulation. Despite increasing concerns regarding the system's security, it undergoes regular audits and testing by representatives from all political parties and various organisations, including cybersecurity experts. So far, there has been no concrete evidence of corruption in the system [1, 3, 50]. There are currently 30 parties registered at the Superior Electoral Court (TSE)[10]. Each party is assigned a unique identification number, which is used as part of a candidate's ID. For most positions (e.g., president and governor), each party can field at most one candidate, and their ID corresponds to the party number itself. Candidate IDs are prominently featured in campaign materials, as voters must type them to cast their e-vote.

Our dataset was compiled using the Twitter streaming API. Data collection commenced in August 2018 and continued until the API's termination in March 2023. We focused on the presidential elections and, for each candidate, monitored (i) the official Twitter account, (ii) the official campaign hashtag (often following the pattern "#<last name>+<candidate ID>"), and (iii) the candidate's full name. We also tracked the Twitter account of the Superior Electoral Court (TSE). Table 1 provides an overview of the keywords employed during the 2018 election cycle.

In July 2022, the TSE officially released the updated list of candidates for the 2022 elections. This event prompted the sole adjustment to the set of keywords over the five-year period. Table 2 features the revised list of candidates and associated keywords. Additionally, we initiated monitoring of the official accounts of Brazilian political parties and the Supreme Court (STF). Table 3 presents the parties' accounts added for the 2022 cycle.

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

**Table 1: 2018 Candidates and Government Accounts Keywords** — This table lists the keywords used in data collection from August 30, 2018, until June 30, 2022. Highlighted terms were retained for the 2022 election cycle. Terms in *italics* are descriptive and were not tracked.

| Name | Account | Hashtag |
|---|---|---|
| Alvaro Dias | @alvarodias_ | #AlvaroDias19 |
| Cabo Daciolo | @CaboDaciolo | #Daciolo51 |
| **Ciro Gomes** | **@cirogomes** | **#Ciro12** |
| **Jose Maria Eymael** | **@Eymaeloficial** | **#Eymael27** |
| Fernando Haddad | @Haddad_Fernando | #Haddad13 |
| Geraldo Alckmin | @geraldoalckmin | #Alckmin45 |
| **Guilherme Boulos** | **@GuilhermeBoulos** | **#Boulos50** |
| Henrique Meirelles | @meirelles | #Meirelles15 |
| **Jair Bolsonaro** | **@jairbolsonaro** | #Bolsonaro17 |
| Joao Amoedo | @joaoamoedonovo | #Amoedo30 |
| Joao Goulart Filho | @joaogoulart54 | #Goulart54 |
| **Luiz Inacio Lula da Silva** | **@LulaOficial** | **#Lula13** |
| Marina Silva | @MarinaSilva | #Marina18 |
| **Vera Lucia** | **@verapstu** | **#Vera16** |
| *Superior Electoral Court* | **@TSEjusbr** | #Eleiçoes2018 |

**Table 2: 2022 Candidates and Government Accounts Keywords** — This table presents the keywords used in data collection from July 1, 2022, until March 14, 2023. Highlighted terms carried over from the 2018 election cycle. Terms in *italics* are descriptive and were not tracked.

| Name | Account | Hashtag |
|---|---|---|
| Andre Janones | @AndreJanonesAdv | #Janones70 |
| **Ciro Gomes** | **@cirogomes** | **#Ciro12** |
| **Jose Maria Eymael** | **@Eymaeloficial** | **#Eymael27** |
| Felipe Avila | @lfdavilaoficial | #Davila30 |
| **Guilherme Boulos** | **@GuilhermeBoulos** | **#Boulos50** |
| **Jair Bolsonaro** | **@jairbolsonaro** | #Bolsonaro22 |
| Leonardo Pericles | @LeoPericlesUP | #Pericles80 |
| Luciano Bivar | @bivaroficial | #Bivar44 |
| **Luiz Inacio Lula da Silva** | **@LulaOficial** | **#Lula13** |
| Pablo Marçal | @pablomarcal | #Marcal90 |
| Simone Tebet | @simonetebetbr | #Tebet15 |
| Sofia Manzano | @SofiaManzanoPCB | #Manzano21 |
| **Vera Lucia** | **@verapstu** | **#Vera16** |
| *Superior Electoral Court* | **@TSEjusbr** | #Eleiçoes2022 |
| *Supreme Court* | @STF_oficial | — |

**Table 3: Tracked Political Parties (2022 Cycle)** — This table lists the Twitter accounts of Brazilian political parties tracked in the data collection from July 1, 2022, to March 14, 2023, with additional information available on the TSE website [10].

| Party | Account |
|---|---|
| *Brazilian Democratic Movement* | @MDB_Nacional |
| *Brazilian Labour Party* | @ptb14 |
| *Democratic Labour Party* | @PDT_Nacional |
| *Workers Party* | @ptbrasil |
| *Comunist Party of Brazil* | @PCdoB_Oficial |
| *Brazilian Socialist Party* | @PSBNacional40 |
| *Brazilian Social Democracy Party* | @PSDBoficial |
| *Christian Social Party* | @pscnacional |
| *Citizenship* | @23cidadania |
| *Green Party* | @partidoverde |
| *Forward* | @somosavante70 |
| *Progressives* | @Progressistas11 |
| *Unified Socialist Workers Party* | @pstu |
| *Brazilian Comunist Party* | @PCBpartidao |
| *Brazilian Labour Renewal Party* | @prtboficial |
| *Party of the Worker's Cause* | @PCO29 |
| *We Can* | @podemos19 |
| *Republicans* | @republicanos10 |
| *Socialism and Freedom Party* | @psol50 |
| *Liberal Party* | @plnacional_ |
| *Democratic Social Party* | @PSD_55 |
| *Social Order Republican Party* | @prosnacional |
| *Solidarity* | @solidariedadeBR |
| *New Party* | @partidonovo30 |
| *Sustainability Network* | @REDE_18 |
| *Popular Unit* | @UP80BR |
| *United Brazil* | @uniaobrasil44 |

5580.

[9] Farhan Asif Chowdhury, Lawrence Allen, Mohammad Yousuf, and Abdullah Mueen. 2020. On Twitter purge: a retrospective analysis of suspended users. In *Companion proceedings of the web conference 2020*. 371–378.

[10] Superior Electoral Court. 2023. Partidos políticos registrados no TSE. https://www.tse.jus.br/partidos/partidos-registrados-no-tse.

[11] Joao MM Couto, Julio CS Reis, Ítalo Cunha, Leandro Araújo, and Fabrício Benevenuto. 2022. Characterizing Low Credibility Websites in Brazil through Computer Networking Attributes. In *2022 IEEE/ACM International Conference on Advances in Social Networks Analysis and Mining (ASONAM)*. IEEE, 42–46.

[12] Stefano Cresci, Roberto Di Pietro, Angelo Spognardi, Maurizio Tesconi, and Marinella Petrocchi. 2023. Demystifying Misconceptions in Social Bots Research. *arXiv preprint arXiv:2303.17251* (2023).

[13] Maureen A. Eger and Sarah Valdez. 2015. Neo-nationalism in Western Europe. *Europ. Sociological Rev.* 31, 1 (2015), 115–130. https://doi.org/10.1093/esr/jcu087

[14] Fatima Ezzeddine, Omran Ayoub, Silvia Giordano, Gianluca Nogara, Ihab Sbeity, Emilio Ferrara, and Luca Luceri. 2023. Exposing influence campaigns in the age of LLMs: a behavioral-based AI approach to detecting state-sponsored trolls. *EPJ Data Science* 12, 1 (2023), 1–21.

[15] Emilio Ferrara. 2017. Disinformation and social bot operations in the run up to the 2017 French presidential election. *arXiv preprint arXiv:1707.00086* (2017).

[16] Eric Ferreira Dos Santos, Danilo Carvalho, Livia Ruback, and Jonice Oliveira. 2019. Uncovering social media bots: a transparency-focused approach. In *Companion Proceedings of The 2019 World Wide Web Conference*. 545–552.

[17] James N Green. 2022. Brazilian Democracy in the Balance: Ahead of the October 2 presidential election, Jair Bolsonaro continues to stoke expectations of fraud, raising fears of a possible January 6-style attack. *NACLA Report on the Americas* 54, 3 (2022), 253–257.

[18] Chris Hays, Zachary Schutzman, Manish Raghavan, Erin Walk, and Philipp Zimmer. 2023. Simplistic Collection and Labeling Practices Limit the Utility of Benchmark Datasets for Twitter Bot Detection. In *Proceedings of the ACM Web Conference 2023*. 3660–3669.

[19] Marc Owen Jones. 2019. The gulf information war| propaganda, fake news, and fake trends: The weaponization of twitter bots in the gulf crisis. *International journal of communication* 13 (2019), 27.

[20] Tobias R Keller and Ulrike Klinger. 2019. Social bots in election campaigns: Theoretical, empirical, and methodological implications. *Political Communication* 36, 1 (2019), 171–189.

[21] Will Knight. 2022. Why it's so hard to count twitter bots? https://www.wired.com/story/twitter-musk-bots/

[22] Luca Luceri, Ashok Deb, Adam Badawy, and Emilio Ferrara. 2019. Red bots do it better: Comparative analysis of social bot partisan behavior. In *Companion proceedings of the 2019 world wide web conference*. 1007–1012.

[23] Luca Luceri, Ashok Deb, Silvia Giordano, and Emilio Ferrara. 2019. Evolution of bot and human behavior during elections. *First Monday* (2019).

[24] Caio Machado, Beatriz Kira, Vidya Narayanan, Bence Kollanyi, and Philip Howard. 2019. A Study of Misinformation in WhatsApp groups with a focus on the Brazilian Presidential Elections.. In *Companion proceedings of the 2019 World Wide Web conference*. 1013–1019.

[25] Enrico Mariconti, Jeremiah Onaolapo, Syed Sharique Ahmad, Nicolas Nikiforou, Manuel Egele, Nick Nikiforakis, and Gianluca Stringhini. 2017. What's in a Name? Understanding Profile Name Reuse on Twitter. In *Proceedings of the 26th International Conference on World Wide Web*. 1161–1170.

[26] Michele Mazza, Marco Avvenuti, Stefano Cresci, and Maurizio Tesconi. 2022. Investigating the difference between trolls, social bots, and humans on Twitter. *Computer Communications* 196 (2022), 23–36.

[27] Innocent Mbona and Jan HP Eloff. 2022. Feature selection using Benford's law to support detection of malicious social media bots. *Information Sciences* 582 (2022), 369–381.

[28] Mehwish Nasim, Andrew Nguyen, Nick Lothian, Robert Cope, and Lewis Mitchell. 2018. Real-time detection of content polluters in partially observable Twitter networks. In *Companion Proceedings of the The Web Conference 2018*. 1331–1339.

[29] Martin N Ndlela. 2020. Social media algorithms, bots and elections in Africa. *Social media and elections in Africa, Volume 1: Theoretical perspectives and election campaigns* (2020), 13–37.

[30] neeljai. [n. d.]. Is there a bug or is this account creation date for real? joined January 1970. https://www.reddit.com/r/Twitter/comments/q8f0ng/is_there_a_bug_or_is_this_account_creation_date/

[31] Diogo Pacheco, Pik-Mai Hui, Christopher Torres-Lugo, Bao Tran Truong, Alessandro Flammini, and Filippo Menczer. 2021. Uncovering coordinated networks on social media: methods and case studies. In *Proceedings of the international AAAI conference on web and social media*, Vol. 15. 455–466.

[32] Javier Pastor-Galindo, Mattia Zago, Pantaleone Nespoli, Sergio López Bernal, Alberto Huertas Celdrán, Manuel Gil Pérez, José A Ruipérez-Valiente, Gregorio Martínez Pérez, and Félix Gómez Mármol. 2020. Spotting political social bots in Twitter: A use case of the 2019 Spanish general election. *IEEE Transactions on Network and Service Management* 17, 4 (2020), 2156–2170.

[33] Francesco Pierri, Alessandro Artoni, and Stefano Ceri. 2020. Investigating Italian disinformation spreading on Twitter in the context of 2019 European elections. *PloS one* 15, 1 (2020), e0227821.

[34] Iacopo Pozzana and Emilio Ferrara. 2020. Measuring bot and human behavioral dynamics. *Frontiers in Physics* (2020), 125.

[35] Tommaso Radicioni, Fabio Saracco, Elena Pavan, and Tiziano Squartini. 2021. Analysing Twitter semantic networks: the case of 2018 Italian elections. *Scientific Reports* 11, 1 (2021), 13207.

[36] Julio CS Reis, Philipe Melo, Kiran Garimella, Jussara M Almeida, Dean Eckles, and Fabrício Benevenuto. 2020. A dataset of fact-checked images shared on whatsapp during the brazilian and indian elections. In *Proceedings of the international AAAI conference on web and social media*, Vol. 14. 903–908.

[37] Julie Ricard and Juliano Medeiros. 2020. Using misinformation as a political weapon: COVID-19 and Bolsonaro in Brazil. *Harvard Kennedy School Misinformation Review* 1, 3 (2020).

[38] Patrícia Rossini, Camila Mont'Alverne, and Antonis Kalogeropoulos. 2023. Explaining beliefs in electoral misinformation in the 2022 Brazilian election: The role of ideology, political trust, social media, and messaging apps. *Harvard Kennedy School Misinformation Review* 4, 3 (2023).

[39] Mauricio Savarese and Diane Jeantet. 2023. Brazil's Jair Bolsonaro is barred from running for office until 2030. https://apnews.com/article/brazil-bolsonaro-ineligible-court-ruling-vote-99dee0fe4b529019ccbb65c9636a9045

[40] Márcio Silva and Fabrício Benevenuto. 2021. COVID-19 ads as political weapon. In *Proceedings of the 36th Annual ACM Symposium on Applied Computing*. 1705–1710.

[41] Richard Takacs and Ian McCulloh. 2019. Dormant bots in social media: Twitter and the 2018 US senate election. In *Proceedings of the 2019 IEEE/ACM International Conference on Advances in Social Networks Analysis and Mining*. 796–800.

[42] Serena Tardelli, Marco Avvenuti, Maurizio Tesconi, and Stefano Cresci. 2020. Characterizing social bots spreading financial disinformation. In *International conference on human-computer interaction*. Springer, 376–392.

[43] Christopher Torres-Lugo, Manita Pote, Alexander C Nwala, and Filippo Menczer. 2022. Manipulating Twitter Through Deletions. In *Proceedings of the International AAAI Conference on Web and Social Media*, Vol. 16. 1029–1039.

[44] Joshua Uyheng and Kathleen M Carley. 2021. Computational analysis of bot activity in the Asia-Pacific: A comparative study of four national elections. In *Proceedings of the international AAAI conference on web and social media*, Vol. 15. 727–738.

[45] Onur Varol, Emilio Ferrara, Clayton Davis, Filippo Menczer, and Alessandro Flammini. 2017. Online human-bot interactions: Detection, estimation, and characterization. In *Proceedings of the international AAAI conference on web and*

social media, Vol. 11. 280–289.

[46] Soroush Vosoughi, Deb Roy, and Sinan Aral. 2018. The spread of true and false news online. *Science* 359, 6380 (2018), 1146–1151. https://doi.org/10.1126/science.aap9559

[47] Kai Cheng Yang, Pik Mai Hui, and Filippo Menczer. 2019. Bot Electioneering Volume: Visualizing social bot activity during elections. In *The Web Conference Companion*. 214–217. https://doi.org/10.1145/3308560.3316499

[48] Kai-Cheng Yang and Filippo Menczer. 2023. Anatomy of an AI-powered malicious social botnet. *arXiv preprint arXiv:2307.16336* (2023).

[49] Kai-Cheng Yang, Onur Varol, Pik-Mai Hui, and Filippo Menczer. 2020. Scalable and Generalizable Social Bot Detection through Data Selection. In *Proc. 34th AAAI Conf. on Artificial Intelligence (AAAI)*.

[50] Cesar Zucco Jr and Jairo M Nicolau. 2016. Trading old errors for new errors? The impact of electronic voting technology on party label votes in Brazil. *Electoral Studies* 43 (2016), 10–20.

