# OpenReview forum: "Bots, Elections, and Controversies: Twitter Insights from Brazil's Polarised Elections"
_ACM.org/TheWebConf/2024/Conference — TheWebConf24 Oral_

### Official Review · Reviewer_7hbR · 2023-11-17

**Novelty:** 2
**Technical Quality:** 6

**Review:**

This paper provides an interesting observational study related to Brazilian political discussions on Twitter. The authors examine a large dataset spanning 5 years of tweets and covering two presidential elections, studying users’ behavioral patterns of activity, creation, and interaction. They study suspicious bot accounts revealing interesting tendencies in the use of replies. The work is relevant to the track and the whole community. The rationale and technicality of the paper are not novel despite the case study being new. ​​The paper is well-written, and the quality of the presentation is more than satisfactory.

My only concern is about the level of contribution provided by this manuscript. Although it is methodologically robust and presents an unseen case study, its novelty and insights are limited. For instance, I do not see how “This work can be instrumental in dismantling coordinated campaigns” given the already-existing methodologies used to discover coordinated actions and bot accounts.

Pros:
1. Robust methodology
2. Interesting and unseen longitudinal case study
3. Quality and clarity of presentation

Cons:
1. Limited originality and technical contribution
2. Results could be expanded to further characterize bots’, suspicious users’, and coordinated accounts’ activity
3. Insights are not that instrumental, as pointed out by the authors

**Questions:**

1. it’s not clear why authors believe users exit the conversation and why they speculate about a “a plausible interpretation” based on “a substantial presence of bots within the Twitter ecosystem.” I’d encourage the authors to elaborate on the rationale of this intuition.
2. the distinction between datasets used in previous research and the one used in this paper is not clear: "This discrepancy can be attributed to the distinctions between our dataset, which is domain-specific, and the datasets employed in prior research, which were domain agnostic."
3. the definition of suspicious users and bot accounts is somewhat unclear, and the two terms are used interchangeably: what’s the reason for introducing the suspicious users group?
4. it would be ideal to show that the “overall stable trend persists” at varying thresholds
5. not clear if fig. 5 is related only to bots or all the users
6. account creation date in correspondence with political events is nothing new: The authors should position their contribution against the extant (large body of) literature.

**Reviewer Confidence:**

4: The reviewer is certain that the evaluation is correct and very familiar with the relevant literature

**Scope:**

4: The work is relevant to the Web and to the track, and is of broad interest to the community

---

### Official Review · Reviewer_w4Tx · 2023-11-22

**Novelty:** 4
**Technical Quality:** 5

**Review:**

This study analyses tweets associated with Brazilian politics during two presidential election cycles.

Positive aspects of the study:
-The theme of the study is relevant and shows results for a phenomenon of a country relatively under-explored in the literature.
-In general, the text is well written.
-Large-scale data from an important phenomenon.


Negative aspects of the study:

-Focus of the paper could be clear. Some analyses are a bit disconnected, lacking a better discussion, for example, the one regarding fig3 – section 2.3. Raising research questions and direct analysis to answer them could help.

-Presentation and organization. The flow of the presentation could be improved. Data/data collection was presented last; this was not the ideal strategy in this specific paper. Also, missed a better discussion of related work – a section of related work could have helped. Some related studies are presented in different parts of the paper; however, a clear contrast between the present study and previous ones should be better introduced. It is important to contextualize other studies on Brazilian elections using social media – in a quick search, I found some potential candidates that should be included.

-Methods. As the collection changes during the analysis, the effect on the results should be better investigated.

Minor issues:

-The legend of fig 1a is hard to understand (bold line legends).
- what is the x-axis of fig1b?

**Questions:**

I am having trouble interpreting the PDF presented in fig 2. Something seems to be off, is it?

Missed better discussions of the implications of the results. Taking previous works into consideration, is the phenomenon observed a surprise? Should we expect something different from what we already know?

Is the phenomenon observed exclusively from the Brazilian scenario?

Does the phenomenon observed only happen in the studied system?

How could the study be important in dismantling coordinated campaigns?

**Ethics Review Description:**

-

**Reviewer Confidence:**

4: The reviewer is certain that the evaluation is correct and very familiar with the relevant literature

**Scope:**

4: The work is relevant to the Web and to the track, and is of broad interest to the community

---

### Official Review · Reviewer_LFTE · 2023-11-23

**Novelty:** 6
**Technical Quality:** 6

**Review:**

The paper is a substantial contribution to understanding the role of social bots in political contexts, particularly in the case of Brazilian elections. It opens avenues for future research and policy-making in the domain of digital democracy and computational propaganda in underscrutinized settings.
 The study's focus on the role of social bots in political discourse is highly pertinent given the increasing influence of social media on politics.
Ideally, a more in-depth analysis of specific bot strategies and their countermeasures would further enhance the paper's impact and relevance.




 Strengths:

Extensive Dataset: The paper analyzes a significant dataset of 437 million tweets generated by 13 million accounts, providing a comprehensive view of Twitter activity related to Brazilian politics between 2018 and 2023​​. To my knowledge, this is the largest dataset ever studied for South America, which is an often understudied domain.

Methodologies: The research employs novel methods for uncovering patterns, such as examining account creation dates to detect coordinated bot activities​​.

Detailed Writing: The paper provides a thorough overview of the Brazilian electoral process, enhancing the contextual understanding of the data​​.

Weaknesses:

Lack of Depth in Some Areas: While the paper covers a broad range of details, certain areas, such as the specific tactics employed by the bots and their exact influence on the discourse, could be explored in more depth.

Potential Biases in Data Collection: The reliance on specific keywords and hashtags for data collection might introduce biases, as it may overlook relevant data not encapsulated by these terms​​. This needs to be further discussed.

Limited Discussion on Countermeasures: The paper could benefit from a more detailed discussion on strategies to counteract malicious bot activities and their implications for social media platforms and political processes.

Update: I ack reading the authors' responses and considered making score changes as deemed appropriate.

**Questions:**

Impact Assessment: How do you assess the direct impact of the detected bot activities on the political discourse and electoral outcomes? Is there a way to quantify their influence?

Algorithmic Bias and Limitations: Could there be limitations or biases in the algorithms used for detecting bots, particularly in differentiating between coordinated human activities and bot operations?

Future Research Directions: The paper mentions future research directions in analyzing account dynamics in political discourse​​. Could you elaborate on the specific methodologies or approaches you envision for this future work?

Broader Implications: How do your findings on Brazilian politics extend to other global political scenarios? Are there universal patterns, or do bot strategies significantly differ across different political and cultural contexts?

**Reviewer Confidence:**

4: The reviewer is certain that the evaluation is correct and very familiar with the relevant literature

**Scope:**

4: The work is relevant to the Web and to the track, and is of broad interest to the community

---

### Official Review · Reviewer_53wZ · 2023-11-23

**Novelty:** 6
**Technical Quality:** 6

**Review:**

This contribution analyzes Twitter data from 2018 to 2023 to detect information manipulation campaigns. The authors detect an increasing presence of bots and evidence of coordination (accounts sharing Twitter handles and creation of many accounts on the same day).

*Strengths:*
- extensive analysis over several years.
- original and significant work.
- analysis is mainly based on proven methodologies.

*Weaknesses:*
- It would have been interesting to combine the different analyses to see if some of the accounts created at the same time are also bots and also the ones who share handles or not.
- I think section 4 should appear before the results.

Just a comment: The account creation date in 1970 is likely a bug, as this corresponds to the UNIX time, which is probably the default value.

**Questions:**

See weaknesses above.

**Ethics Review Description:**

no issues

**Reviewer Confidence:**

3: The reviewer is confident but not certain that the evaluation is correct

**Scope:**

4: The work is relevant to the Web and to the track, and is of broad interest to the community

---

### Official Review · Reviewer_wC9s · 2023-11-24

**Novelty:** 5
**Technical Quality:** 5

**Review:**

Pros
* The theoretical motivation of the paper is properly discussed, and the focus on Brazil is effectively justified
* I appreciate the comprehensiveness of the analyses encompassing basic descriptions, bots' engagement patterns (retweets vs. replies, for instance), coordinated accounts, and new born users

Cons
* Rather than reasons to reject this paper, I want to see more analysis on coordinated accounts: 1) Is it possible to see the timeline of the prevalence of coordinated accounts (including the proportion of these accounts/networks relative to the entire data) 2) How do the networks evolve over time, in terms of density, clusters, etc?

**Questions:**

See above

**Ethics Review Description:**

-

**Reviewer Confidence:**

4: The reviewer is certain that the evaluation is correct and very familiar with the relevant literature

**Scope:**

3: The work is somewhat relevant to the Web and to the track, and is of narrow interest to a sub-community

---

### Decision · Program_Chairs · 2024-01-22

**Decision:**

Accept (Oral)

**Comment:**

Quoting Reviewer 7hbR: "This paper provides an interesting observational study related to Brazilian political discussions on Twitter. The authors examine a large dataset spanning 5 years of tweets and covering two presidential elections, studying users' behavioral patterns of activity, creation, and interaction. They study suspicious bot accounts revealing interesting tendencies in the use of replies."

 The reviewers appreciated the chosen topic and saw the clear relevance to the Web community and, in particular to the track. The overall presentation quality is good, though certain elements and figures went through rounds of clarification in the discussion. The technical quality is good-to-excellent and, certainly, rated well above average for the submitted papers.

 The only point of contention IMHO relates to the assessment of the novelty of the work as there is little-to-no methodological contribution. As the authors write (in the discussion) their objective "was to investigate whether the suspicious behaviours observed in other contexts were replicated in [their] specific dataset". Given the scale of the analysis and the importance of the topic studied, I see this as an acceptable contribution.

 Thanks to the engaged discussion by the reviewers, the paper will likely improve further for future/camera-ready versions.